# Reducing Panic Buying During Crisis Lockdowns: A Randomized Controlled Trial of a Theory-Based Online Intervention

**DOI:** 10.3390/bs16010042

**Published:** 2025-12-24

**Authors:** Karina T. Rune, Trent N. Davis, Jacob J. Keech

**Affiliations:** 1School of Health, University of the Sunshine Coast, Sippy Downs, QLD 4556, Australia; tdavis1@usc.edu.au; 2School of Applied Psychology, Griffith University, Brisbane, QLD 4111, Australia

**Keywords:** panic buying, integrated social cognition, consumer behavior, crisis communication, COVID-19, intervention

## Abstract

COVID-19 lockdown announcements triggered global waves of panic buying, leading to widespread panic buying of essential goods and supply chain disruptions. Although the acute phase of the pandemic has passed, panic buying continues to emerge during natural disasters, extreme weather events, and other crisis-related disruptions, highlighting the ongoing need for evidence-based strategies to address its psychological drivers. Social cognition constructs, including willingness, intentions, attitudes, subjective norms, and risk perceptions, have been identified as modifiable psychological predictors of panic buying. However, few studies have experimentally tested theory-driven interventions aimed at modifying these mechanisms. This study evaluated the effectiveness of a brief, online intervention based on integrated social cognition models in reducing panic-buying-related cognitions during a hypothetical lockdown scenario. A pre-registered randomized controlled trial was conducted with Australian grocery shoppers (*N* = 140), who were randomly allocated to an intervention or control condition. Participants completed self-report measures assessing their willingness, intentions, attitudes, subjective norms, and risk perceptions at both pre- and post-intervention times. The hypotheses were partially supported. Compared with the control condition, the intervention group reported greater reductions across targeted psychological constructs. For hygiene products, significant decreases were observed across all five constructs, and for non-perishable foods, willingness, intention, and attitudes significantly decreased. For cleaning products, reductions were evident for attitudes, subjective norms, and intentions. These findings suggest that theory-informed, scalable interventions can effectively modify the social cognition processes underlying panic buying. This study extends existing research and demonstrates the potential for brief, theory-based communication strategies to reduce panic-buying-related cognitions. Future research should evaluate these interventions in real-world settings and explore mechanisms to target automatic cognitive processes.

## 1. Introduction

The Severe Acute Respiratory Syndrome Coronavirus 2 (COVID-19) was declared a global pandemic in March 2020 ([59]). As of March 2025, COVID-19 has resulted in nearly 778 million cases and over 7 million deaths worldwide ([60]). In Australia, strict border controls initially limited the number of cases; however, the emergence of highly transmissible variants, such as Delta and Omicron, drove successive waves of infection ([7]; [8], [9]). Public health measures, including lockdowns, travel restrictions, and vaccination campaigns, reduced transmission in the short term but also generated widespread psychological stress and behavioral responses, including panic buying ([12]; [29]; [32]).

Panic buying is a recurrent behavioral response observed during crises such as pandemics, natural disasters, economic instability, and geopolitical conflict (e.g., [18]; [24]; [57]). It is typically defined as the rapid and excessive purchasing of essential household items in anticipation of future scarcity. Panic buying can destabilize supply chains, prompt retailers to impose quotas, and disproportionately affect vulnerable populations, including those from lower socioeconomic backgrounds ([15]; [39]). Panic buying often coincides with the announcement of restrictions. [29] ([29]) found that lockdown announcements across 54 countries triggered short bursts of consumer panic, typically lasting 7–10 days. Beyond creating immediate shortages, panic buying also heightens community anxiety and erodes public trust. An Australian natural experiment also linked lockdowns to poorer mental health, especially for women with young children and those living in constrained housing ([12]), underscoring how stress and perceived loss of control can fuel panic buying.

Although panic buying and hoarding both involve the accumulation of goods, they are conceptually distinct. Panic buying is situational, reactive, and socially contagious, arising from acute events and perceptions of scarcity ([63]). In contrast, hoarding is chronic and deliberate behavior, linked to enduring traits such as intolerance of uncertainty, perfectionism, and difficulty discarding possessions ([22]; [50]). Empirical evidence reinforces these differences. [17] ([17]) compared predictors of panic buying and hoarding in samples from Australia and the US (*N* = 359). They found that panic buying was predicted by perceived scarcity and situational anxiety specific to COVID-19, whereas hoarding was linked to general intolerance of uncertainty and prior compulsive behaviors. [55] ([55]) also distinguished rational stockpiling motives (e.g., low food reserves, concerns about infection) from irrational panic buying, which was tied to poor mood and a herd mentality. These findings suggest that panic buying is a short-lived coping response to situational stress and social contagion, rather than enduring pathology.

A growing body of research highlights panic buying as a complex behavioral response shaped by multiple influences. [37] ([37]) identified six categories of influences: cognitive, emotional, social, economic, behavioral, and government-related factors. [63] ([63]) highlighted four psychological processes underpinning panic buying: perceived threat and scarcity, fear of the unknown, coping motives, and social contagion. Together, these reviews demonstrate that panic buying is not attributable to a single cause but emerges from overlapping psychological and social processes. Recent empirical studies demonstrate how these factors are manifested during crises. [13] ([13]) showed that scarcity cues intensified both cognitive (e.g., competitiveness) and emotional (e.g., anxiety) responses, driving stockpiling, particularly when trust in government is low. [61] ([61]) found that media influence, psychological panic, group norms and identity increased the likelihood of panic buying, whereas stronger social control reduced it. [39] ([39]) further demonstrated how Australian news coverage during COVID-19 disproportionately blamed individual shoppers, particularly those from lower socioeconomic and ethnically diverse backgrounds, while portraying supermarkets as both “victims” and “heroes.” This framing reinforced stigma and obscured systemic contributors such as supply chain fragility. Collectively, these findings suggest that crisis communication strategies risk amplifying stigma and mistrust unless they deliberately target social cognitive processes, including perceived norms, and collective efficacy and responsibility.

Research on demographic predictors of panic buying has yielded inconsistent results. [10] ([10]) reported that household income and the presence of children predicted greater purchasing, while [19] ([19]) found that household size was the only significant factor, with gender, age, and education unrelated. [62] ([62]) observed panic buying across income classes, whereas [45] ([45]) emphasized that psychological traits, such as intolerance of uncertainty, were stronger predictors of panic buying than demographics. Similarly, [43] ([43]) found no significant effects of income, household size, or gender in an Australian sample. Taken together, these findings suggest that demographic characteristics may shape vulnerability in some contexts, but they are inconsistent and less reliable than psychological processes in explaining panic buying.

### 1.1. Current Mitigation Strategies

Efforts to curb panic buying during the COVID-19 pandemic and prior crises have relied mainly on structural and informational measures, such as purchase limits, rationing, and government messaging ([28]). These approaches can provide short-term relief but often fail to address underlying psychological drivers and may inadvertently worsen the problem. For example, [40] ([40]) found that retailer purchase limits sometimes intensified panic buying by signaling anticipated scarcity. Similarly, [45] ([45]) reported that higher perceived threat reduced shopping frequency, but increased quantities purchased, particularly when media emphasized shortages. Further, [23] ([23]) showed in a simulated experiment, combined strategies integrating supply monitoring, government response, and psychological counseling were more effective than supply monitoring alone. These findings highlight that structural controls alone are insufficient and risk backfiring without parallel strategies that directly address the social and psychological mechanisms sustaining panic buying.

Public messaging shows similarly mixed outcomes. [14] ([14]), analyzing 80,000 Weibo comments (China’s largest social media platform), found that messages emphasizing material sufficiency from trusted authorities were most effective in calming panic buying, whereas vague and non-mandatory appeals had little impact. [20] ([20]) compared three government message strategies (i.e., norms and reciprocity, scarcity reassurance and moral appeals), and found that moral appeals (e.g., obligations to protect vulnerable groups) reduced both intentions and simulated panic buying behavior, while scarcity reassurance had minimal effect. [15] ([15]) also showed that panic buying was driven by perceived scarcity and cues to action, suggesting that messaging should minimize scarcity cues and leverage social norms. [51] ([51]) experimentally demonstrated that even in stable supply chains, scarcity cues prompted hoarding and phantom orders, underscoring the influence of situational stressors and emotions. Together, these findings suggest that effective interventions must target cognitive and social drivers, particularly perceived norms, moral obligations, and scarcity beliefs, rather than relying solely on structural or informational measures. However, given the limitations of research to date, no consensus “state-of-the-art” mitigation strategy currently exists.

### 1.2. Theoretical Frameworks

While the reviewed studies provide valuable insights into panic buying, a systematic review by [11] ([11]) highlighted that panic buying research remains fragmented and under-theorized, with limited attention to cognitive and affective mechanisms. Applying theory to identify modifiable constructs provides a structure for designing effective interventions. Theoretical approaches are also advantageous because framework-based messages are generally more effective and easier to evaluate than those developed without a theoretical foundation ([56]). Social cognition models have a long tradition of successfully explaining and changing health-related behavior ([26]). This makes them especially relevant for panic buying, which is driven by both social influence and individual appraisal, and therefore requires an approach that can systematically target modifiable psychological constructs.

The theory of planned behavior (TPB; [2]), has been widely applied to predict a range of health-related behaviors. Within the TPB, behavioral intention is shaped by attitudes toward the behavior, subjective norms, and perceived behavioral control. Several studies have applied the TPB to panic-buying contexts. [42] ([42]), surveying Romanian consumers (*N* = 518) during the COVID-19 lockdown, found that negative attitudes and descriptive social norms predicted intentions to stockpile and actual purchasing, whereas perceived behavioral control was non-significant. Similarly, [53] ([53]) reported that attitudes and subjective norms predicted panic buying among Malaysian consumers, while perceived behavioral control showed limited predictive power. Together, these findings suggest that in uncertain contexts, attitudes and social influence outweigh perceived behavioral control in driving panic buying.

While TPB constructs are relevant, the model has well-recognized limitations. It struggles to explain the intention–behavior gap, where individuals fail to act on intentions in real-world contexts ([38]). It also emphasizes deliberative processes, overlooking the impulsive, emotionally driven responses that shape many health behaviors, including panic buying. Moreover, much variance in behavior remains unexplained by TPB constructs. These limitations align with conceptual analyses of panic buying during COVID-19 that emphasize affective and automatic processes. [5] ([5]) argued that panic buying is driven by perceived scarcity, loss of control, anticipated regret, and herd behavior, reflecting a coping response to uncertainty and fear. In their systematic review, [11] ([11]) concluded that models like the TPB cannot fully explain panic buying, which often occurs under emotional arousal. This underscores the need for integrative approaches that combine deliberative and impulsive pathways and extend the TPB with constructs such as risk perception and social influence.

To address these gaps, researchers have extended social cognition models with constructs that capture reactive and affective pathways. These include behavioral willingness and impulsivity, drawn from dual-process theories that highlight the role of fast, emotion-driven responses alongside deliberative reasoning ([21]; [52]). Another extension is risk perception, referring to individuals’ subjective sense of personal vulnerability if they do or do not engage in a behavior, which is central to health frameworks like the health belief model ([41]) and the health action process approach ([47]).

In panic-buying contexts, affective risk perceptions and behavioral willingness may explain variance beyond traditional TPB constructs. [34] ([34]) demonstrated that panic buying during COVID-19 was driven by a cognition–affect pathway, where scarcity reduced perceived control and heightened panic, leading to impulsive purchasing. Likewise, [33] ([33]) used a dual-system stimulus–organism–response framework to show that social norms and affective reactions (fear, uncertainty) predicted panic buying alongside reflective appraisals of control. [31] ([31]) also found that attitudes toward stockpiling, social norms, and fear of future unavailability were the strongest predictors of panic buying in Germany. Those who stockpiled did so mainly due to uncertainty, fear of shortages, and a desire to reduce shopping frequency, whereas those who refrained cited trust in supply chains and altruistic motives. These findings emphasize that panic buying is driven both by deliberative (attitudes, control) and automatic/affective (fear, scarcity cues) pathways.

[43] ([43]) extended this evidence in Australia, showing that attitudes, subjective norms, and risk perceptions best predicted purchasing across food, hygiene, and cleaning products, while automaticity and individual differences were weaker. Importantly, predictors varied by product type, underscoring the need for category-specific interventions. Experimental evidence also supports targeting social cognition directly. [20] ([20]) showed that moral appeals and normative prompts reduced simulated panic buying, whereas reassurance-based messages had little effect. Collectively, this research provides support for brief, scalable interventions grounded in integrated social cognition frameworks.

The present study builds on this literature by testing an integrated social cognition framework through a brief video-based intervention. Drawing on evidence-based behavior change methods ([30]), the intervention targeted attitudes, subjective norms, willingness, intentions, and risk perceptions. By testing this intervention experimentally, the study extends prior work beyond descriptive accounts to evaluate whether modifying these processes reduces panic-buying cognitions across product categories in a simulated crisis. The central research question was: Can a brief, theory-informed intervention reduce willingness and intention to panic buy, as well as shifts in attitudes, subjective norms, and risk perceptions? In line with prior evidence highlighting the importance of these constructs, we hypothesized that:

### 1.3. Primary Outcomes

Participants in the intervention condition would report significantly lower (1) willingness and (2) intention to increase purchasing of (a) non-perishable products, (b) hygiene products, and (c) cleaning products in the hypothetical lockdown scenario from pre- to post-intervention, compared to the control group.

### 1.4. Secondary Outcomes

Participants in the intervention condition would report significantly lower (1) attitudes, (2) subjective norms, and (3) risk perceptions regarding increased purchasing of (a) non-perishable products, (b) hygiene products, and (c) cleaning products in the hypothetical lockdown scenario from pre- to post-intervention, compared to the control group.

## 2. Method

### 2.1. Participants

A convenience sample of Australian community members (*N* = 140) was recruited through multiple channels, including paid Facebook advertisements, online message boards, and email invitations sent to university staff. Appendix A presents the CONSORT flow diagram outlining participant recruitment, randomization, exclusions, and final sample allocation. Eligible participants were required to: (1) be aged 18 years or older, (2) currently reside in Australia, and (3) self-identify as someone who regularly purchases groceries. Participants’ ages ranged from 24 to 88 years (*M* = 49.37, *SD* = 15.47). The sample was predominantly female (70.7%), with 54% reporting they had children, and 58% holding a bachelor’s degree or higher qualification. This demographic profile is consistent with prior research indicating that women and individuals with higher education levels are more likely to participate in online psychological studies (e.g., [49]). Although the study employed a convenience sample, this was considered appropriate for an experimental design focused on testing causal psychological mechanisms rather than estimating population prevalence. Additional demographic details are presented in Table 1.

### 2.2. Materials

#### 2.2.1. Intervention Development

The intervention content was developed using evidence-based behavior change methods ([30]), mapped to theoretical constructs derived from integrated social cognition models (see Table 2). The video aimed to modify social cognitive predictors of panic buying, including attitudes, subjective norms, intentions, willingness, and perceived risk. Development was informed by prior research identifying these constructs as key psychological drivers of panic buying during crises ([43]). The selection of self-affirmation, risk framing, and perspective-taking was guided by [30]’s ([30]) taxonomy of behavior change methods. Self-affirmation reduces defensiveness and promotes alignment with personal values, enhancing receptivity to normative and risk-based messages. Risk framing increases perceived vulnerability and consequence awareness, which is critical for motivating behavior change in crisis contexts. Perspective-taking fosters empathy and social reflection, supporting shifts in subjective norms and attitudes. Additional elements included environmental re-evaluation (highlighting how panic buying affects others) and normative information (emphasizing the approval of family and friends).

Prior to the trial, a small pilot study (*N* = 22, *M*age = 32.75, *SD* = 8.77) was conducted to assess the clarity, believability, and practical feasibility of the video content. The aim was to ensure that the materials were understandable and acceptable to participants, rather than to validate effects across demographic subgroups. Feedback was collected via open-ended questions. Participants’ responses indicated that the content was clear, believable, and realistic. Minor refinements were incorporated before the pre-registered main study. Full intervention materials are available via the Open Science Framework.

#### 2.2.2. Intervention Condition

Participants in the intervention condition viewed a four-minute narrated video presented as a PowerPoint with voiceover. The narrator adopted a calm and empathic tone, guiding participants through the content in a conversational style. Visuals were simple and text-based, designed to maintain focus and encourage emotional engagement. The video revisited the hypothetical COVID-19 lockdown scenario described earlier in the survey and presented scenario-based risk information about how buying more than necessary can contribute to shortages. Environmental re-evaluation was prompted through examples of affected groups, including healthcare workers, first responders, vulnerable individuals, and carers. Normative prompts suggested that family and friends would want the viewer to buy only what they need. Participants were also invited to take the perspective of a frontline worker unable to access essentials after a shift, encouraged to personalize risk by reflecting on their own role in preventing shortages, and completed a self-affirmation exercise. These elements were designed to enhance emotional engagement and promote reflection on the personal and social consequences of panic buying, while deliberately avoiding content that may induce fear or distress.

#### 2.2.3. Control Condition

Participants in the control condition completed a three-minute neutral mental imagery task (“tangy lemon”; [27]). The task involved imagining the sensory experience of interacting with a tangy lemon and did not include any content related to panic buying or behavior change constructs. It served as an active attention control, matching the intervention for duration and cognitive engagement without introducing panic-buying or behavior-change content. Although the task did not equate for emotional or social-reflective engagement, this conservative design increases confidence that any observed effects can be attributed to the theory-based elements of the intervention.

#### 2.2.4. Randomization

Participants were randomly allocated to either the intervention or control group using Qualtrics’ built-in randomization feature. The platform employs a Mersenne Twister pseudorandom number generator seeded with a Unix timestamp (in milliseconds) to ensure allocation randomness. Randomization occurred immediately after the completion of pre-intervention measures.

### 2.3. Measures

All measures were adapted based on validated procedures and standard guidelines ([1]; [3]), which helped to ensure construct validity. In addition, these theory-based measures adapted to the current context were previously used by [43] ([43]) to measure panic buying social cognitions, and factorial validity was established using structural equation modeling. Constructs assessed included willingness, intentions, attitudes, subjective norms, and risk perceptions, each measured separately for three product categories: (a) non-perishable foods, (b) hygiene products, and (c) cleaning supplies. In line with dual-process models of health behavior ([25]), willingness and intention were conceptualized as distinct constructs. Intention reflects a deliberate, reasoned commitment to perform a behavior, typically formed through conscious evaluation of attitudes, norms, and perceived behavioral control. Willingness, by contrast, reflects a more reactive, spontaneous openness to engage in a behavior when prompted by situational cues or social context. This distinction is particularly relevant in crisis contexts, where behavior may be shaped by immediate social or emotional triggers rather than by planned decision-making.

The survey items were answered with reference to a standardized hypothetical COVID-19 lockdown scenario, which asked participants to imagine a future lockdown marked by media reports of panic buying. This scenario was used solely as a reference for participant responses, not as content within the intervention video itself.

“It has just been announced by your state premier that your state will be re-entering lockdown in a few days due to a new outbreak of COVID-19. This means that you can only leave your house for essential reasons such as going to work or buying essential items. It is not yet known how long the lockdown will be imposed for. The media has also started to report that people have been “panic buying” and that supermarkets are running out of hygiene products such as toilet paper, non-perishable foods such as pasta and tinned vegetables, and cleaning products such as wipes and hand sanitizer.” Participants completed pre- and post-intervention versions of each measure. All items used a 7-point Likert scale or semantic differential response scale. All psychological measures demonstrated good internal consistency, with Cronbach’s α values ranging from 0.81 to 0.91 (Willingness = 0.87–0.91; Intention = 0.86–0.90; Attitudes = 0.81–0.85; Subjective Norms = 0.90–0.91; Risk Perception = 0.88–0.90). Full reliability statistics for each construct by product category are reported in Appendix A. Behavioral proxies were not included due to practical and ethical constraints during the early stages of the COVID-19 pandemic, when future behaviors and lockdown conditions were unpredictable. Instead, the study focused on validated self-report scales to examine social cognition predictors of panic buying. Item wording is detailed in the Survey Measures Table Appendix A.

### 2.4. Data Quality Checks

Two attention-check items (e.g., “please choose option ‘disagree’ to ensure you are paying attention”) were included to screen for inattentive responding ([35]; [46]). Participants who failed one or more attention checks (*N* = 49) were excluded from the data analysis. Although this represents a relatively high exclusion rate (35%), attention checks were deliberately stringent to maximize data quality and minimize the impact of inattentive responding. No additional manipulation checks of message comprehension were included; however, the high completion rate and consistent intervention effects across constructs provide indirect evidence of participant engagement and message alignment. This procedure was also systematic and pre-registered.

### 2.5. Procedure

Following ethics approval from the University Human Research Ethics Committee (S201470), the study was administered online via Qualtrics. After providing informed consent, participants completed a demographic questionnaire and the pre-intervention measures. They were then randomly assigned to either the intervention or control condition. Participants in the intervention group viewed a four-minute theory-based video, while those in the control group completed a neutral three-minute mental imagery task. Following this, all participants repeated the same set of survey measures. The full study procedure took approximately 25 min to complete. The pre–post design enabled comparison of outcomes across time and experimental condition to evaluate the effectiveness of the intervention. The study was preregistered on the Open Science Framework prior to data collection: https://osf.io/9jdbm/ (accessed on 17 November 2025).

### 2.6. Data Analysis

The study employed a randomized controlled trial design with a two-group mixed model structure. The independent variables were Time (pre-intervention, post-intervention; within-subjects) and Group (intervention, control; between-subjects). The dependent variables were willingness, intentions, attitudes, subjective norms, and risk perceptions, each assessed separately for three product categories: non-perishable foods, hygiene products, and cleaning supplies. A series of 2 × 2 mixed-model ANOVAs were conducted to examine Time × Group interaction effects. Where significant interactions were found, these were followed by simple effects analyses to examine within-group and between-group differences over time. Assumptions of the mixed-model ANOVAs were examined prior to analysis. Most variables met normality criteria, although willingness (T1, T2) and attitude (T1) showed mild deviations. Data transformations were conducted but produced no differences in statistical inferences; therefore, untransformed results are reported for clarity and interpretability. All statistical tests used a conservative alpha level of α = 0.01 to control for Type I error.

## 3. Results

### 3.1. Primary Outcomes

The intervention had a significant effect on willingness and intention to increase purchasing behavior in the hypothetical scenario for some product categories (see Figure 1).

#### 3.1.1. Non-Perishable Food Products

A mixed model ANOVA revealed a significant Time × Group interaction for intention (*F*(1,138) = 7.87, *p* = 0.006), indicating a differential change over time between the intervention and control groups. However, the Time × Group interaction for willingness (*F*(1,138) = 6.37, *p* = 0.013) fell just short of our specified cutoff for significance. Turning to simple effects analysis, post-intervention, willingness (*p* = 0.003) and intention (*p* = 0.001) significantly decreased in the intervention group from pre- to post-intervention. Additionally, post-intervention intention was significantly lower in the intervention group compared to the control group (*p* = 0.008). No other effects met the reporting threshold of *p* < 0.01.

#### 3.1.2. Cleaning Products

A mixed model ANOVA revealed a significant Time × Group interaction for intention (*F*(1,138) = 6.74, *p* = 0.010), indicating that changes over time differed between groups. Simple effects, post-intervention, found that intention (*p* = 0.004) significantly decreased in the intervention group from pre- to post-intervention. Additionally, post-intervention intention was significantly lower in the intervention group compared to the control group (*p* = 0.004). No other effects met the reporting threshold of *p* < 0.01.

#### 3.1.3. Hygiene Products

A mixed model ANOVA revealed a significant Time × Group interaction for willingness (*F*(1,138) = 6.98, *p* < 0.001) and intention (*F*(1,138) = 8.26, *p* = 0.005), indicating a differential change over time between the intervention and control groups. Simple effects analysis, post-intervention, revealed that willingness (*p* < 0.001) and intention (*p* < 0.001) significantly decreased in the intervention group from pre- to post-intervention. However, post-intervention scores were not significantly different between the intervention and control groups (*p*s > 0.01).

### 3.2. Secondary Outcomes

The intervention also significantly impacted attitudes, subjective norms, and risk perception to panic buy for some specific product categories (see Figure 2).

#### 3.2.1. Attitudes

For non-perishable food products, a mixed model ANOVA revealed a significant Time × Group interaction for attitudes (*F*(1,138) = 16.57, *p* < 0.001), with attitudes significantly decreasing from pre- to post-intervention in the intervention group (*p* < 0.001). For hygiene products, the Time × Group interaction was also significant (*F*(1,138) = 24.28, *p* < 0.001), with attitudes significantly decreasing from pre- to post-intervention in the intervention group (*p* < 0.001). Similarly, for cleaning products, a significant Time × Group interaction was found (*F*(1,138) = 25.05, *p* < 0.001), with attitudes again significantly decreasing in the intervention group over time (*p* < 0.001). No post-intervention differences between the intervention and control groups met the reporting threshold of *p* < 0.01.

#### 3.2.2. Subjective Norms

For hygiene products, a mixed model ANOVA revealed a significant Time × Group interaction (*F*(1,138) = 15.48, *p* < 0.001), with subjective norms significantly decreasing in the intervention group from pre- to post-intervention (*p* < 0.001). A similar pattern was observed for cleaning products (*F*(1,138) = 13.52, *p* < 0.001), with a significant pre- to post-intervention reduction in the intervention group (*p* < 0.001). For non-perishable food products, the Time × Group interaction did not meet the reporting threshold (*F*(1,138) = 5.54, *p* = 0.020), and although a pre–post reduction in subjective norms was observed in the intervention group (*p* < 0.001), this result is not interpreted further in the absence of a significant interaction. No post-intervention between-group differences met the reporting threshold of *p* < 0.01 and are therefore not reported.

#### 3.2.3. Perceived Risk

For hygiene products, a mixed model ANOVA revealed a significant Time × Group interaction for perceived risk (*F*(1,138) = 8.55, *p* = 0.004), with a significant reduction observed in the intervention group from pre- to post-intervention (*p* < 0.001). For non-perishable food products and cleaning products, the Time × Group interactions did not meet the reporting threshold (*F*(1,138) = 4.94, *p* = 0.028; *F*(1,138) = 5.69, *p* = 0.018, respectively) and are therefore not interpreted further. Although both categories showed significant reductions in perceived risk within the intervention group from pre- to post-intervention (*p* < 0.01), no between-group differences at post-intervention reached significance (*ps* > 0.01).

## 4. Discussion

This study evaluated the impact of a brief intervention on psychological drivers of panic buying across three product categories: non-perishable, hygiene, and cleaning products. The primary hypotheses were partially supported across product categories. For non-perishable products, intention showed a significant interaction effect, with reductions evident both within the intervention group and between groups at post-intervention. Willingness did not meet the significance threshold for interaction (*p* = 0.01), though a significant within-group reduction was observed. For cleaning products, intention showed a significant interaction effect, with reductions in both within-group and between-group comparisons, while willingness showed no significant effects. For hygiene products, both willingness and intention showed significant interaction effects, though reductions were confined to within-group differences, with no between-group effects.

The secondary hypotheses were also partially supported. For non-perishable products, attitudes showed a significant interaction effect with reductions limited to the intervention group. Subjective norms and risk perceptions did not interact significantly but both reduced significantly within the intervention group. For hygiene products, attitudes, subjective norms, and risk perceptions all showed significant interactions and within-group reductions, but no between-group differences. Cleaning products followed a similar pattern: attitudes and subjective norms interacted significantly with within-group reductions, while risk perceptions decreased only within the intervention group and did not reach significance for the interaction. These patterns align with integrated social cognition models, which emphasize both reflective constructs (attitudes, norms, intentions) and impulsive processes (risk perceptions). The stronger and more consistent effects for hygiene products may reflect their heightened emotional salience and perceived urgency during health crises, underscoring the need to tailor interventions to engage both cognitive and affective pathways.

This study contributes to the growing body of research on panic buying during global crises ([44]; [48]; [58]). Extending prior research identifying these as key predictors of panic buying ([43]), the intervention produced reductions across all five targeted constructs for hygiene products, although these effects were largely confined to within-group changes. From an integrated social cognition perspective, these findings highlight the influence of both reflective and more automatic responses in driving panic buying behavior. Hygiene products may be particularly influenced by affective processes during health-related crises, given their association with disease prevention. These processes include risk perception and are reinforced by social and psychological pressures such as fear of scarcity and observational learning from others’ stockpiling ([36]; [63]). The within-group reductions in subjective norms and risk perceptions suggest that the intervention may have disrupted socially reinforced and risk-related drivers of panic buying, consistent with theoretical predictions. These findings also align with the existing research emphasizing the central role of social influence and perceived scarcity in driving the hoarding of essential items ([5]; [28]), and support the utility of integrated social cognition models for designing interventions that target multiple cognitive pathways underlying consumer behavior in crisis contexts.

For non-perishable food items, the intervention reduced intentions and produced within-group reductions in attitudes, constructs central to the deliberate planning and the perceived utility of these products during crises ([36]; [54]). This partially aligns with previous research identifying attitudes, subjective norms, and risk perceptions as key predictors of non-perishable stockpiling during the COVID-19 pandemic ([43]). However, the absence of significant interaction effects for willingness and perceived risk in the current study suggests that automatic, fear-driven responses linked to survival preparedness may be less amenable to brief interventions. Non-perishables are frequently prioritized during crises due to their long shelf life and role in preparedness, reinforcing a sense of urgency and necessity ([39]; [54]). The lack of significant effects for willingness and risk perceptions highlights the challenge of addressing panic buying for essential food items. These constructs, shaped by ingrained survival instincts, may require more intensive interventions, such as messaging that communicates supply stability and builds trust in logistics systems, to reduce these entrenched psychological factors underpinning panic buying ([37]; [54]).

The effect of the intervention on social cognition related to cleaning products was mixed. The intervention influenced social and attitudinal determinants (i.e., intention, attitudes, and subjective norms), while constructs such as willingness and perceived risk were less responsive. This divergence suggests that while deliberate and normative drivers of cleaning-product stockpiling may be amenable to brief interventions, more automatic or affectively charged responses are harder to shift. Cleaning products are often perceived as supplementary rather than essential during crises ([15]), which may reduce their perceived urgency and intervention impact. This pattern is consistent with [51] ([51]), who reported that environmental stress can lead to irrational purchasing, though such responses vary depending on a product’s perceived relevance to crisis preparedness.

### Theoretical Implications

The findings of this study provide insights into the application of integrated social cognition models for designing messaging aimed at reducing panic buying during health and environmental crises. These models incorporate both reflective processes and more automatic or affective processes, offering a comprehensive framework for understanding behavior change. In the current study, the intervention had the strongest and most consistent effects on reflective constructs of attitudes, subjective norms, and intentions, indicating that these social cognitions are amenable to change in the context of panic buying behavior. While the study did not assess actual behavior, changes in these constructs are often viewed as proximal indicators of behavioral readiness in integrated models. For example, reductions in attitudes toward panic buying, particularly for hygiene and non-perishable products, suggest that beliefs about necessity and urgency may be modifiable through brief interventions ([43]; [63]). Reductions in subjective norms also point to the potential for messaging strategies to counteract social cues that reinforce stockpiling tendencies, although the study did not test whether norms predict behavior directly. Similarly, intentions to panic buy were effectively reduced across most product types, suggesting the intervention may impact motivational precursors of behavior.

In contrast, willingness and risk perceptions showed more variability, particularly for non-perishable food items and cleaning products supporting the view that automatic, emotionally charged responses are harder to shift ([51]; [54]). This pattern likely reflects differences in cognitive processing: attitudes and intentions are shaped by deliberative pathways and respond to structured informational interventions ([2]; [26]), whereas willingness and risk perception are driven by automatic, affective, or situational cues ([25]; [52]). The brief, cognitively oriented format of the video appears to be well-suited to reflective constructs but perhaps less effective at targeting impulsive or emotionally charged responses. This is consistent with dual-process models (e.g., [52]), which argue that interventions targeting automatic processes must engage affective systems more directly; for example, through vivid imagery, narrative storytelling, or simulated crisis experiences. Future research should examine whether tailoring intervention components to the cognitive–affective profile of each construct enhances intervention effectiveness.

A key strength of this study is its theory-driven design, which provided a structured framework for targeting panic buying social cognitions using mapped behavior change methods ([30]). The use of evidence-based behavior change methods that have been used across a broad range of contexts provides some confidence that the intervention is likely to be scalable and could be considered for use in future crises. However, ongoing evaluation of such programs is essential. A further strength of the study is that we included multiple product types which enabled a more nuanced understanding of how psychological responses vary depending on the perceived urgency or function of different items.

Despite these strengths, several limitations should be noted. The central limitation is reliance on a hypothetical lockdown scenario and self-reported measures. While vignette-based methods are widely used in experimental designs to test psychological mechanisms and can enhance external validity ([4]; [6]), they cannot fully capture the urgency, stress, and scarcity cues that accompany real-world crises. Similarly, self-report measures provide theoretically relevant insights but may not always reflect actual consumer purchasing behavior. This raises questions about ecological validity and it is recommended that more objective measures of behavior be considered when evaluating any future real-world applications.

The brief nature of the intervention may have constrained its impact, particularly for constructs like willingness and risk perception, which may require more emotionally engaging strategies. Although a conservative significance threshold (*p* = 0.01) was used to minimize Type I error, the study may have been underpowered to detect smaller effects, raising the possibility of Type II error. Moreover, we did not examine participant-level moderators (e.g., age, gender, income, household size, or prior experiences of scarcity). This was consistent with our preregistered focus on product-level effects, but it represents an important limitation: demographic and contextual factors may shape both the salience of panic buying and the effectiveness of interventions, and future studies should incorporate these analyses. Finally, females and those with higher education levels were also overrepresented, and therefore, results should be generalized with this in mind.

Finally, while the brief, video-based format of the intervention may help to facilitate scalability due to low cost, reach, and ease of dissemination, real-world implementation during crises is likely to encounter additional barriers. These include media saturation, conflicting public messages, and fluctuating levels of institutional trust, all of which can dilute or override intervention effects ([16]; [39]). In acute crises, individuals are often influenced more by emotionally charged or sensationalist content than by structured behavioral messaging. The scalability of this intervention should therefore be seen as a strength of its format, but not as a guarantee of effectiveness across all crisis contexts.

Importantly, panic buying does not occur in a psychological vacuum; it is shaped by broader contextual factors such as media narratives, government messaging, and perceived supply chain reliability. Future studies should incorporate behavioral proxies, such as simulated shopping tasks, virtual store environments, or purchasing data, to test whether changes in social cognition translate into actual consumer behavior. Second, research is needed to examine how contextual factors, including media narratives, government messaging, and perceived supply chain reliability, interact with intervention effects. Third, identifying individual-level moderators (e.g., prior experiences of scarcity, household size, or income) will be critical for tailoring the intervention to different consumer groups. Finally, future research should expand the application of integrated social cognition models to other crises, such as natural disasters, to evaluate broader utility and inform preparedness planning. Together, these steps will help clarify the boundary conditions of the intervention and inform whether, how, and for whom it can be scaled effectively in crisis settings.

## 5. Conclusions

In conclusion, this study provides preliminary evidence that a brief, theory-informed intervention can reduce psychological processes underpinning panic buying. The strongest effects were observed for hygiene products, highlighting the value of targeting both reflective and impulsive processes. At the same time, limitations related to the use of hypothetical scenarios, self-report measures, and the absence of behavioral outcomes must be addressed. Future work should prioritize behavioral validation, examine contextual moderators, and refine intervention strategies before considering large-scale implementation. Overall, the results suggest that theory-informed communication strategies show promise for mitigating panic buying during crises, but further work is needed to ensure effectiveness and scalability in real-world settings.

## Figures and Tables

**Figure 1 behavsci-16-00042-f001:**
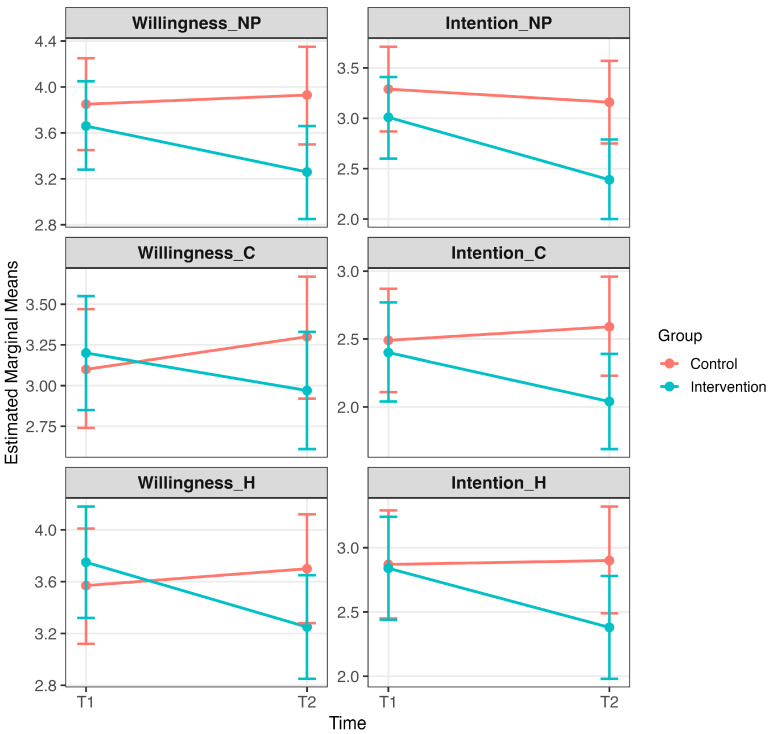
Primary outcomes: Effects of the intervention on willingness and intention to panic buy. Note. Estimated marginal means (±95% CI) are shown for willingness to purchase non-perishable products (NP), willingness to purchase cleaning products (C), and intention to purchase hygiene products (H). Means are displayed separately for intervention and control groups at pre-intervention (T1) and post-intervention (T2). Significant Time × Group interactions and/or pre–post reductions within the intervention group indicate that the intervention reduced willingness and intention to panic buy in some product categories (*p* < 0.01). Full descriptive statistics and estimated marginal means are provided in Appendix A.

**Figure 2 behavsci-16-00042-f002:**
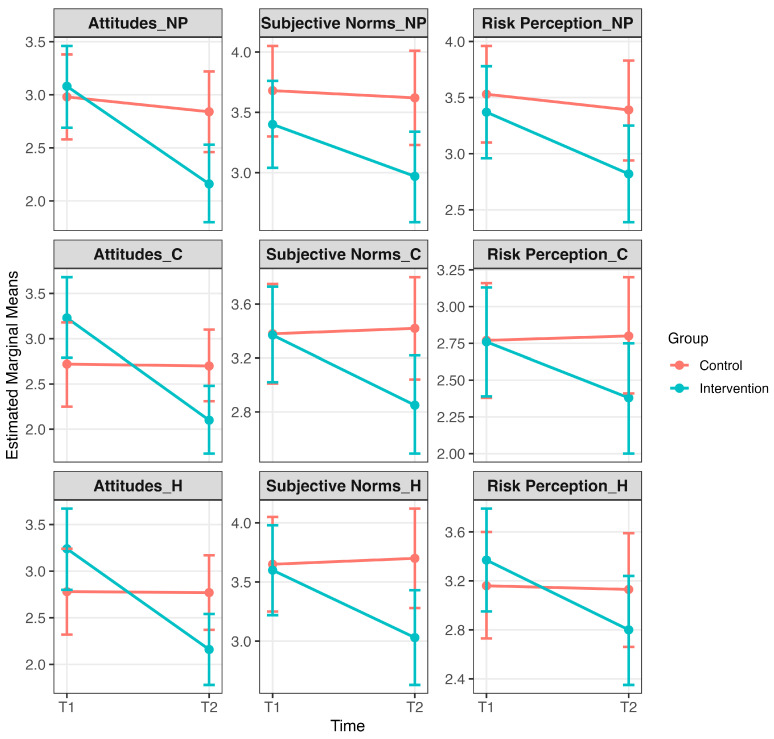
Secondary outcomes: Effects of the intervention on attitudes, subjective norms, and perceived risk related to panic buying. Note. Estimated marginal means (±95% CI) are shown separately for intervention and control groups at pre-intervention (T1) and post-intervention (T2). Panels display attitudes toward non-perishable products (NP), attitudes toward cleaning products (C), attitudes toward hygiene products (H), subjective norms for non-perishable products (NP), subjective norms for cleaning products (C), subjective norms for hygiene products (H), perceived risk related to non-perishable products (NP), perceived risk related to cleaning products (C), and perceived risk related to hygiene products (H). Significant Time × Group interactions and/or pre–post reductions within the intervention group indicate that the intervention reduced social cognitive drivers of panic buying in several domains (*p* < 0.01). Full descriptive statistics and estimated marginal means are provided in Appendix A.

**Table 1 behavsci-16-00042-t001:** Demographic Characteristics of Participants by Condition.

Characteristic	Control Group		Intervention Group	
	*n*	%	*n*	%
**Sex**				
Male	16	23.9	19	26.0
Female	50	74.6	49	67.1
Different identity	1	1.5	5	6.8
**Marital Status**				
Partnered (married or de facto)	43	64.2	34	46.6
In relationship	8	11.9	13	17.8
Separated, divorced or widowed	6	9.0	16	21.9
Single	10	14.9	10	13.7
**Children in Household**	39	58.2	37	50.7
**Employment**				
Employed (full- or part-time)	40	59.7	39	53.4
Home duties or unemployed	17	25.4	23	31.5
Student (full- or part-time)	8	11.9	11	15.1
**Education Level**				
University	44	65.7	37	50.7
TAFE/trade/VET	11	16.4	24	32.9
High school or less	12	17.9	12	16.4
**Weekly Work Hours**				
0–19 h	11	16.4	10	13.7
20–39 h	23	34.4	18	24.6
40+ h	21	31.3	10	13.7
No response	12	17.9	35	47.9
**Annual Household Income**				
<$41,600	16	23.9	17	23.3
$41,600–$77,999	19	28.3	25	34.3
≥$78,000	26	38.8	27	37.0
No response	6	9.0	4	5.5

**Table 2 behavsci-16-00042-t002:** Behavior Change Methods Mapped to Theoretical Constructs.

Behavior Change Method	Implementation Strategy	Targeted Construct(s)
Scenario-based risk information	Revisited the hypothetical lockdown scenario and provided information about how excessive purchasing can lead to shortages.	Risk perception, attitude
Environmental re-evaluation	Described how panic buying affects others, including healthcare workers, first responders, older adults, and carers.	Attitude, subjective norm
Information about others’ approval	Stated that family and friends would want participants to only buy what they need.	Subjective norm
Shifting perspective	Invited participants to imagine being a frontline worker unable to access groceries after a shift.	Risk perception, attitude, intention
Personalized risk feedback	Prompted participants to consider the role they can play in preventing shortages.	Risk perception, intention
Self-affirmation	Asked participants to reflect on whether panic buying is consistent with their values.	Attitude, intention

## Data Availability

The materials, data, syntax, and output presented in this study are openly available on the Open Science Framework (OSF) at https://osf.io/2kd9a/ (accessed 17 November 2025).

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
