# Peer review of "Reducing Panic Buying During Crisis Lockdowns: A Randomized Controlled Trial of a Theory-Based Online Intervention"

_behavsci, 2025, doi:10.3390/bs16010042_

Round 1
Reviewer 1 Report
Comments and Suggestions for Authors
This study presents a well-designed, theory-driven intervention aimed at reducing panic buying behaviors during hypothetical COVID-19 lockdowns. The manuscript is clearly written, methodologically sound, and addresses an important public health issue. The findings contribute to the literature on crisis-related consumer behavior and offer practical insights for policymakers. However, the study would benefit from a detailed literature review, clarifications regarding intervention scalability, inclusion of behavioral outcomes, and limitations due to its hypothetical setting.
The article needs to be rewritten and have its structure reorganized to meet the journal's guidelines available at: https://www.mdpi.com/journal/behavsci/instructions
Title: Could specify the study design (e.g., "A Randomized Controlled Trial") for methodological transparency.
Abstract: The abstract should be a total of about 200 words maximum. The abstract does not clarify that the outcomes assessed were psychological constructs (e.g., attitudes, intentions), not observed behavior. Hypotheses are listed but not explicitly tied to results (e.g., "supported/partially supported").
Introduction: There is no clearly articulated research question(s).
The introduction lacks a critical review of prior interventions targeting panic buying or related crisis behaviors. While theoretical frameworks (e.g., TPB) are discussed, the authors fail to: (1) examine existing mitigation strategies (e.g., purchase limits, public messaging), (2) compare panic buying to similar behaviors (e.g., hoarding), or (3) clearly identify gaps justifying this intervention. This omission weakens the study's novelty and practical relevance. A concise paragraph summarizing past approaches and their limitations would strengthen the rationale for this theory-based solution.
Excessive citation stacking without analytical synthesis. Many references are listed sequentially without integration or evaluation. Suggestion: split the section and create a Literature Review (or Theoretical Background) section. The section should be divided into two distinct parts: an Introduction and a Literature Review.
Redundant mentions of "integrated… models" without advancing new insights.
Missing Section: Theoretical Framework: A key weakness of the manuscript is the absence of a dedicated literature review section. Although several relevant studies are cited throughout the introduction, there is no structured synthesis of prior findings on panic buying, its psychological and socioeconomic predictors, or previously tested interventions. The state-of-the-art was not presented.
For instance, the study collects data on participants’ income but does not examine how income level and age may influence panic buying behavior, despite evidence in the literature highlighting its relevance to purchasing power and vulnerability during crises. The lack of such discussion limits the theoretical depth and weakens the contextual framing of the study’s contributions. A clearly defined literature review (or Theoretical Framework) would enhance the manuscript’s academic rigor and position it more effectively within existing research.
Method:
Demographics: Overrepresentation of women (70.7%) and educated participants limits generalizability. No justification for this skew is provided..
High exclusion rate (n = 49, about 35%) due to failed attention checks raises concerns about participant engagement or the clarity of instructions.
The main description of the video intervention is overly condensed, relying too heavily on supplementary materials, which hinders full assessment by reviewers. Video Content: Lacks detail on narrator tone/visuals (critical for emotional engagement).
It is unclear whether the intervention was piloted or validated with diverse demographic groups.
No reliability statistics (e.g., Cronbach’s alpha) are reported for the constructs measured, limiting confidence in internal consistency.
Please clarify the distinction between "willingness" and "intention".
While some variables violated normality assumptions, the manuscript does not fully explain how this was handled or its implications.
Results
Visuals are helpful, but figure captions lack sufficient context for standalone interpretation.
Figures 1–2: poor design: lack of granularity (e.g., error bars overlap; no raw data points).
No discussion of potential Type II errors or limited statistical power in non-significant findings
Discussion
The study's central limitation, reliance on hypothetical scenarios and self-report, is addressed only late in the section and not with sufficient emphasis.
The discussion sometimes over-interprets small or borderline effects, especially in cases where between-group differences were non-significant.
Differences across product types are discussed insightfully but lack integration with participant-level variables (e.g., demographics, past experiences). Was there some analysis about toilet paper?
Mechanisms Unclear: Why did some constructs (e.g., attitudes) change more than others? Needs deeper theorizing.
Overgeneralization: Claims about "scalability" without addressing real-world barriers (e.g., media noise during crises).
Future Directions: Vague suggestions (e.g., "test in real-world settings"); propose specific designs (e.g., partner with supermarkets).
Conclusion: Missing Section
References
Numerous formatting errors in DOIs, including redundant “https://doi.org/” and broken links.
Some references appear duplicated or improperly formatted.
The manuscript should undergo careful citation style compliance check against MDPI standards.
Language and grammar:
The manuscript is generally clear. Some grammatical errors and typos (e.g., “vido[e]”, “attnetion”). Repetition of terms and inconsistent verb tense affect flow and clarity. Technical jargon is used without sufficient explanation.
AI review or plug ins like Grammarly are recommended.
Self citations:
Rune & Keech (2023) – Cited several times throughout the introduction, theoretical framework, methods, and discussion. This prior work appears to be foundational to the current study's theoretical and empirical framing. Line 300: 2013 ou 2023?.
Hagger, Keech et al. (2021), Keech et al. (2021), Keech et al. (2018), Hamilton, Keech, Hagger (2020) are cited in discussions on integrated models and physical distancing but not related to panic buyng.. Some of them seems unnecessary. Please, reduce self citations.
Note:
The reviewer requests that changes in new versions of this manuscript be clearly identified using Word Track Changes or another method of identification (such as highlighting in another color).
Comments on the Quality of English LanguageThe manuscript is generally clear. Some grammatical errors and typos (e.g., “vido[e]”, “attnetion”). Repetition of terms and inconsistent verb tense affect flow and clarity. Technical jargon is used without sufficient explanation.
AI review or plug-ins like Grammarly are recommended.
Reviewer 2 Report
Comments and Suggestions for Authors
Dear authors,
thank you for the opportunity to review your interesting manuscript on panic buying. Please find below my comments. I hope they are helpful.
The introduction is focused on panic buying during the time period of the coronavirus pandemic. The work is reasonably timely and relevant to humanity/ many societies across the world. We are now living endemically, but still the introduction is well motivated. It is easy follow along provides definition and examples and then outlines why panic buying is relevant in other scenarios and explains behavioural and psychological aspects behind it. That is well done
While a wide variety of literature is presented, the evaluation of previous approaches -e.g., Theory of Planned Behaviour by itself fall short
In addition, a literature review on theory-informed behavioural intervention grounded in integrated social cognition principles and evidence-based behaviour change methods are completely missing.
It remains unclear how the hypotheses are grounded within the body of literature. In essence where is framework or the specific literature that underpins the hypotheses of primary and secondary outcomes
A convenience sample is the foundation of the study. While recruitment, selection is clear, there is no justification for the appropriateness of the approach.
The intervention builds on 4-minute video. Can additional information about content tone, visuals and delivery be provided, that complement the table and give more insight.
The rational for the chosen combinations of techniques : self-affirmation, risk framing, perspective-taking should be further elaborated and explained. Justification is missing
Piolet study: Feedback was gathered only through open-ended items, with no Likert scales on engagement, emotional impact, or clarity. The authors indicate that it was "clear, believable, and realistic". Yet there is no evidence examples or studies which underpin this
Control Condition
While the neutral lemon imagery task serves as control with respect to attention. Can the authors provide explanation how this controls for emotional engagement and social reflection which are part of the intervention? If there is failure to control- how can we be sure that theory specific effects are reflected. I rather say emotional priming.
Measurement
The self-reported constructs are fine, but why were no behavioural proxies included into the work?
Where are measures of construct validity and internal consistency of scales? Please report them
Manipulation Check
Have manipulation checks been conducted? There is no information. Please provide details. If not, please justify. How do we know otherwise that participants were attentive with respect to the video, understood the intervention messages, and ultimately their responses were aligned with the intervention goals
Comments on the Quality of English LanguageI am a non-native speaker, I can't judge.
Round 2
Reviewer 1 Report
Comments and Suggestions for Authors
A theoretical framework section was added, and citations are somewhat better integrated. However, prior intervention strategies (e.g., quotas, messaging, structural controls) are not critically reviewed. No clear paragraph on state-of-the-art mitigation strategies.
Limitations mentioned but not sufficiently emphasized; some claims remain overstated; no analysis of demographic moderators; mechanistic explanations still weak; scalability still overgeneralized; future directions remain vague.
Author Response
Comment 1: A theoretical framework section was added, and citations are somewhat better integrated. However, prior intervention strategies (e.g., quotas, messaging, structural controls) are not critically reviewed. No clear paragraph on state-of-the-art mitigation strategies.
Response 1: We thank the reviewer for this comment. In our previous revision we addressed this point by including two dedicated paragraphs reviewing prior intervention strategies (e.g., quotas, rationing, and messaging approaches). To make this clearer in the current version, we have now introduced a specific heading for this section, which explicitly signals that it provides a critical review of mitigation strategies. We have also refined our critique to emphasise that, while quotas, messaging campaigns, and structural controls have been trialled, these approaches have limitations and no consensus “state-of-the-art” mitigation strategy currently exists.
Please see track change page 5 and 6 of the revised manuscript.
Comment 2: Limitations mentioned but not sufficiently emphasized; some claims remain overstated; no analysis of demographic moderators; mechanistic explanations still weak; scalability still overgeneralized; future directions remain vague.
Response 2: We thank the reviewer for this thoughtful feedback. In response, we carefully reviewed and expanded the limitations section to ensure it is comprehensive. We also examined the entire discussion to tone down the strength of claims and ensure they are fully aligned with the data presented. With regard to demographic moderators, we did not conduct additional analyses because prior research has found weak or no associations between demographic factors and this behaviour. Moreover, participants were randomly allocated to groups in our experimental design, and given the lack of prior evidence of demographic correlates, we did not have preregistered hypotheses to support such analyses. We can confirm that this rationale has been clarified in the manuscript. We have also tempered our discussion of scalability to avoid overgeneralisation and provided a clearer articulation of future research directions.
Please see track change page 23 and 24 of the revised manuscript.